# Single Cell Determination of 7,8-dihydro-8-oxo-2′-deoxyguanosine by Fluorescence Techniques: Antibody vs. Avidin Labeling

**DOI:** 10.3390/molecules28114326

**Published:** 2023-05-25

**Authors:** Giusy Maraventano, Giulio Ticli, Ornella Cazzalini, Lucia A. Stivala, Mariella Ramos-Gonzalez, José-Luis Rodríguez, Ennio Prosperi

**Affiliations:** 1Istituto di Genetica Molecolare “Luigi Luca Cavalli-Sforza”, CNR, 27100 Pavia, Italy; giusy.maraventano97@gmail.com (G.M.); giulio.ticli@igm.cnr.it (G.T.); 2Dipartimento di Medicina Molecolare, Università di Pavia, 27100 Pavia, Italy; ornella.cazzalini@unipv.it (O.C.); luciaanna.stivala@unipv.it (L.A.S.); 3Zootecnia and Animal Production Laboratory, Faculty of Veterinary Medicine, Major National University of San Marcos, Lima 15081, Peru; mramosgo@unmsm.edu.pe (M.R.-G.); josero05@ucm.es (J.-L.R.); 4Faculty of Veterinary, Universidad Complutense de Madrid, 28040 Madrid, Spain

**Keywords:** 7,8-dihydro-8-oxo-2′-deoxyguanosine, oxidative DNA damage, immunofluorescence, avidin binding

## Abstract

An important biomarker of oxidative damage in cellular DNA is the formation of 7,8-dihydro-8-oxo-2′-deoxyguanosine (8-oxodG). Although several methods are available for the biochemical analysis of this molecule, its determination at the single cell level may provide significant advantages when investigating the influence of cell heterogeneity and cell type in the DNA damage response. to. For this purpose, antibodies recognizing 8-oxodG are available; however, detection with the glycoprotein avidin has also been proposed because of a structural similarity between its natural ligand biotin and 8-oxodG. Whether the two procedures are equivalent in terms of reliability and sensitivity is not clear. In this study, we compared the immunofluorescence determination of 8-oxodG in cellular DNA using the monoclonal antibody N45.1 and labeling using avidin conjugated with the fluorochrome Alexa Fluor488 (AF488). Oxidative DNA damage was induced in different cell types by treatment with potassium bromate (KBrO_3_), a chemical inducer of reactive oxygen species (ROS). By using increasing concentrations of KBrO_3_, as well as different reaction conditions, our results indicate that the monoclonal antibody N45.1 provides a specificity of 8-oxodG labeling greater than that attained with avidin-AF488. These findings suggest that immunofluorescence techniques are best suited to the in situ analysis of 8-oxodG as a biomarker of oxidative DNA damage.

## 1. Introduction

The production of oxidative DNA damage may occur in dependence of several factors, including endogenous mechanisms of generation of reactive oxygen species (ROS) at the level of mitochondrial and/or cellular metabolism. In addition, DNA damage may arise from exogenous cellular factors, such the exposure to chemicals and radiation [1,2]. One of the endpoints of this type of DNA damage is the production of oxidized bases, among which the formation of 7,8-dihydro-8-oxoguanine (8-oxoG) and 7,8-dihydro-8-oxo-2′-deoxyguanosine (8-oxodG) is the most abundant modification induced at the cellular and nuclear level because of the low redox potential of guanine [3,4]. The oxidative DNA damage is typically repaired by the base excision repair (BER) system, and removal of these lesions is critical for avoiding the accumulation of mutations and for preventing genome instability which may lead to carcinogenesis [5,6,7,8].

However, determination of oxidative DNA damage is not only relevant for tumor-related studies, since more recently the association of these lesions with other pathologies, such as neurodegenerative diseases [9,10,11], as well as inflammatory [12] and cardiovascular diseases [13], have been described. Therefore, detecting the cellular levels of oxidative DNA damage, and in particular the determination of 8-oxoG and 8-oxodG, has progressively acquired the importance of a significant biomarker which is useful in diagnostic studies [14,15].

Several methods for the determination of 8-oxodG are in use, and assays have been developed in recent decades ranging from HPLC-based electrochemicals to mass spectrometry techniques [15,16,17,18,19,20]. Although these methodologies are sophisticated and very sensitive, they require the biochemical extraction of DNA which is subject to possible oxidation artifacts [21]. In addition, the relationship between cell type and cell/tissue localization is lost in these procedures. Immunocyto/histochemistry, as well as immunofluorescence techniques, have been developed with the production of specific antibodies, which have been used in a number of studies [22,23,24,25,26,27]. In addition, a cellular assay based on the binding properties exhibited by the glycoprotein avidin towards 8-oxoG/dG was proposed several years ago [28]. This procedure exploits the structural similarity between 8-oxoG/dG and biotin (Figure 1), which is the natural high-affinity ligand of avidin [29]. 

Some works have applied the latter technique using a fluorochrome-labeled avidin for the fluorescence determination of 8-oxodG as a marker of oxidative DNA damage [30,31,32]. Fluorescence-based detection techniques are generally sensitive and allow good spatial localization of the target for the in situ microscopy analysis. However, no information enabling the choice between the immunolabeling, and the ligand technique is available for this type of study. Here, we have compared these methods on different cell types after induction of oxidative DNA damage with potassium bromate (KBrO_3_), a chemical compound which is a well-known inducer of ROS [33,34].

## 2. Results

In order to compare these two methods for the single cell determination of 8-oxodG, different cell types were treated with a relatively high concentration (40 mM) of KBrO_3_ for a short period of time (30 min) to ensure a sufficient sensitivity while avoiding long-term effects of cell death induced by this toxic compound [35]. These conditions were confirmed to produce ROS, as assessed by the DCFH-DA assay (Appendix A).

Since both procedures may detect 8-oxoG in RNA [23,24,25,26], a digestion with RNase A was recommended in previous works [25,26] and was therefore applied in our study related to DNA damage. Both the immunofluorescence technique and the avidin-based assay were applied to the human keratinocyte cell line HaCaT, to primary cultures of normal fibroblasts (LF-1), and to the HeLa cancer cell line. Figure 2 shows fluorescence images obtained from parallel samples of untreated and KBrO_3_-treated HaCaT cells, which were processed for immunofluorescence staining with N45.1 antibody (A) or for direct incubation with Alexa Fluor488 (AF488)-conjugated avidin (B). The results showed that both procedures provided a clear nuclear staining in KBrO_3_-treated samples; however, a faint, yet detectable nuclear fluorescence was also observed in the untreated control cells incubated with avidin-AF488 while only background levels were observed in the samples stained with the antibody. The different extent of labeling was confirmed by the quantification of fluorescence intensity with Image J software (Figure 2C,D). A similar trend of labeling, providing a brighter signal for avidin compared with the antibody, was found in LF-1 human fibroblasts and in HeLa cells (Appendix A).

The detection of 8-oxodG with an antibody relies on the previous denaturation of DNA to give access to the epitope which is masked within the double helix [25,26,27]. In contrast, the procedure using avidin did not report any requirement concerning the accessibility of 8-oxodG [28]. Therefore, we asked whether opening the double helix would allow an increase in the detection of the oxidized base with the avidin-based method. In parallel, we also investigated the influence of omitting DNA denaturation in the antibody-based procedure. As expected, omission of acidic DNA denaturation (-HCl) in the antibody-based procedure reduced the fluorescent signals of both untreated and KBrO_3_ treated cells (Figure 3A) to background levels. In striking contrast, the previous DNA denaturation (+HCl) resulted in a significant increase in the avidin-AF488 fluorescence intensity, although it occurred both in untreated and KBrO_3_-treated samples (Figure 3B). The quantification of fluorescence intensity signals of these samples indicated that DNA denaturation enhanced the binding of avidin-AF488 to nuclear DNA by aproximately 3.5–4.4 times in both untreated and KBrO_3_-treated samples (Figure 3C).

To further understand whether the fluorescence intensity obtained using the antibody or the avidin detection method was proportional to the amount of the oxidative lesions, cells were treated with increasing concentrations of KBrO_3_. Examples of the images obtained and the quantification of the fluorescence intensity of HaCaT cells stained with the antibody N45.1 or with avidin-AF488 are reported in Appendix A This analysis supported the previous data indicating that labeling with avidin-AF488 provided a higher signal not only in KBrO_3_-treated cells but also in untreated control cells. Therefore, in order to compare the results obtained with avidin labeling with those provided by the antibody independently of the absolute value of fluorescence intensity, the mean value of each experiment was normalized to its respective untreated control. In this way, the increase in fluorescence signal as a function of the KBrO_3_ concentration was independent of the actual fluorescence intensity value. The ratio of fluorescence intensity values of treated vs. untreated samples in repeated experiments was quantified and reported in Figure 4A. The results indicate that the fluorescence intensity signals in cells stained with the N45.1 antibody increased linearly with a slope almost 10-fold greater (0.065) than that observed in samples stained with avidin-AF488 (0.007). To further confirm these results in a different cell type, the fluorescence intensity ratio between treated and untreated samples calculated in HaCaT cells treated with 40 mM KBrO_3_ (Figure 4B) was compared with the ratio measured in LF-1 fibroblasts after treatment with a similar concentration (Figure 4C). The results showed that the fluorescence intensity ratio in HaCaT cells was significantly higher (more than 2 times) for the detection with N45.1 antibody than with the avidin method. A similar trend was observed for LF-1 fibroblasts, indicating that the treated/untreated ratio was greater when 8-oxodG was detected with N45.1 antibody rather than with avidin-AF488, independently of the cell type.

Next, we wanted to further evaluate whether the detection of 8-oxodG in nuclear DNA was influenced by a different accessibility to the reagents used in the two procedures. To this end, untreated and KBrO_3_-treated HaCaT cells were embedded in agarose and exposed to high salt extraction in order to obtained nucleoids, i.e., nuclei devoid of most nuclear proteins, such as those used for the comet assay [36,37]. The extent of 8-oxodG labeling obtained with N45.1 antibody or with avidin-AF488 is shown in Figure 5A,B, respectively. It can be observed that the fluorescence intensity images related to the untreated cells showed higher signals when labeled with avidin AF488 than with N45.1 antibody, similar to the results found for the fixed cell samples. The application of the immunofluorescence technique on nucleoids prepared from other cell types was assessed in lymphoblastoid cells (Appendix A) and in the SH-SY5Y neuroblastoma cell line (see below). The results indicate that the procedure based on the antibody allows the reliable detection of 8-oxodG even in other cell types. 

Given that the preparation of nucleoids is the basis of the comet assay, we investigated the possibility to extend our results and perform 8-oxodG labeling after single cell electrophoresis. In the comet assay, the procedure applied to investigate oxidative DNA damage, and in particular the presence of 8-oxodG, makes use of specific enzymes such as formamidopyrimidine DNA glycosylase (Fpg) that will cut and release the altered base. The generated DNA fragments will be then detected by the comet formation [36]. The combination of the two techniques may allow the determination of residual 8-oxodG after the enzymatic digestion. To test this possibility, cell nucleoids obtained from KBrO_3_-treated SH-SY5Y cells were processed for partial Fpg digestion before the alkaline comet procedure. After that, the reaction with the antibody N45.1 was performed. Figure 6 shows the labeling of 8-oxodG with N45.1 antibody in nucleoids not processed for the glycosylase reaction and in a comet formed after partial digestion with Fpg. The results showed that some 8-oxodG could be detected in the head of the comet, and some residual labeling was also present in the tail, thus demonstrating the feasibility of this application.

## 3. Discussion

Even if in situ procedures for the determination of 8-oxodG are less sensitive and provide semi-quantitative information compared with other techniques [15,16,17,18], they are essential for studying the intracellular localization of 8-oxodG and for the analysis of DNA damage in different cell or tissue types. In addition, single cell determination is the right choice when cell abundance is a limiting factor. 

In this work we compared the single cell detection of 8-oxodG with an immunofluorescence procedure [25,26,27] vs. the use of avidin as a specific ligand, since it was applied in other works [28,30,31,32]. In general, the fluorescence intensity signal provided by avidin-AF488 was higher than that obtained with the immunofluorescence procedure, even if for the latter we used an amplification step. Two possible explanations may account for this difference: (i) compared with avidin, the antibody has a lower accessibility to 8-oxodG into DNA; or (ii) avidin binds to DNA with minor specificity, possibly recognizing dG, as previously suggested [38]. The second explanation is supported by the experimental evidence that DNA denaturation resulted in a higher fluorescence intensity than that observed in non-denatured samples, even in untreated control cells. Furthermore, the FI ratio of the signal between treated and untreated cells was clearly lower after labeling with avidin compared with N45.1 antibody-labeled samples. Another indication is the approximately 10 times lower slope provided by avidin when fluorescence intensity signals were correlated with increasing concentrations of KBrO_3_, suggesting that a labeling plateau was reached in these samples. These results suggest that, at least when applied to single cell staining, avidin may also bind dG, thus decreasing the specificity of 8-oxodG detection. In fact, structural studies revealed that avidin binds dG and 8-oxodG in the micromolar range with a difference in K_D_ between the natural and the oxidized base of only two-fold, while biotin is bound in the nanomolar range [38]. In contrast, the study describing the specificity of N45.1 antibody to the oxidized base reported no significant cross-reactivity with the four deoxyribonucleosides, nor with other guanosine modified forms. Furthermore, competition with free 8-oxodG was obtained with a concentration two-orders of magnitude higher using an ELISA test [24].

It is important to note that the difference in the extent of labeling between avidin and N45.1 antibody was independent of the cell type, as HeLa cells also exhibited the same behavior shown by keratinocytes and fibroblasts. These results further support our findings and suggest that avidin probably detects other nucleotides in addition to 8-oxodG.

## 4. Materials and Methods

### 4.1. Cells and Reagents

The human immortalized keratinocyte cell line HaCaT was obtained from IZLER (Brescia, Italy) and was grown in DMEM high-glucose medium (Euroclone, Pero, Italy) supplemented with 10% fetal bovine serum (FBS) (Euroclone), 2% streptomycin/penicillin (Euroclone), and 2% L-glutamine (Euroclone). LF-1 human normal embryonic fibroblasts (from J. Sedivy, Brown University, Providence, RI, USA) were grown in MEM (Euroclone) supplemented with 10% FBS (Gibco, Thermo Fisher, Milan, Italy), 1% streptomycin/penicillin (Euroclone), and 1% L-glutamine (Euroclone). HeLa cells (from ATCC, Manassas, VA, USA) were grown in DMEM medium supplemented with 10% FBS (Euroclone) and 1% streptomycin/penicillin. Lymphoblastoid cells were obtained from C. Baldo (Telethon-Galliera Genetic Biobank, Genova, Italy) and grown in RPMI medium with 10% FBS. The neuroblastoma cell line SH-SY5Y (from ATCC) was grown in DMEM-F12 (1:1) medium supplemented with heat-inactivated 10% FBS, 100 units/mL penicillin, and 100 mg/mL streptomycin. The cells were cultured under sterile conditions and kept in an incubator at 37 °C with a percentage of CO_2_ equal to 5%. Cells were cultured in plastic flasks or on coverslips in 35 mm petri dishes (Sarstedt Italy, Trezzano S/N, Italy) when used for single cell determination.

All chemicals were obtained from Sigma-Aldrich (Merck, Milan, Italy). Potassium bromate stock solution was dissolved in bi-distilled H_2_O at 500 mM and diluted in PBS containing 20 mM Hepes [34]. Deferoxamine mesylate (DFX) was used in certain procedures to reduce artefactual DNA oxidation [20]. DFX was directly dissolved in the relevant extraction buffer solution. Low gelling agarose was dissolved in PBS/cell mixture at the final concentration of 0.8%. Hoechst 33,258 stock solution was prepared by dissolving 1 mg/mL dye in bi-distilled H_2_O.

The anti-8-oxodG mouse monoclonal antibody N45.1 was obtained from JalCa (Shizuoka, Japan); avidin Alexa Fluor 488 conjugate (avidin-AF488, Invitrogen) was obtained from Thermo Fisher. AF488-conjugated secondary antibodies (specified below) were obtained from ThermoFisher, AbCam (Cambridge, UK), or Immunological Sciences (Rome, Italy).

### 4.2. Cell Treatments 

Cells grown on coverslips were treated with a standard concentration of 40 mM potassium bromate (KBrO_3_) in PBS/20 mM Hepes for 30 min, as described [35]. Untreated controls were incubated only in PBS/Hepes. Afterwards, a 20-min recovery was carried out with the specific medium for each cell type. In some experiments, HaCaT cells were treated with KBrO_3_ concentrations ranging from 10 to 40 mM with the same incubation and recovery times above indicated. At the end of treatment, cells were fixed in cold (−20 °C) methanol/acetone mixture (1:1) and samples stored at −20 °C until use [39].

Before proceeding with the labeling reactions (both antibody or avidin), the samples were re-hydrated in PBS for 5 min and then digested with RNase A (100 μg/mL in PBS containing 1 mM EDTA) for 30 min at room temperature (RT).

### 4.3. Nucleoid Preparation

HaCaT cells were mechanically detached, harvested, and centrifuged at 300× *g* for 5 min. After the supernatant was removed, the pellet was washed in PBS and then centrifuged at 300× *g* for 5 min. Subsequently, low gelling 0.8% agarose was added to the pellet and 2 drops were separately placed on a microscope slide previously pre-coated with 1% agarose. Then, coverslips were used to cover each drop to form the gel, and the slides were placed at 4 °C for 20 min. The slides were covered with lysis buffer (2.5 M NaCl, 0.1 M EDTA, 1% Triton, 10 mM Tris, pH 10.0) containing 1.25 mM DFX and kept at 4 °C for 1 h. Subsequently, the slides were placed in denaturation buffer (0.3 M NaOH, 1 mM EDTA) for 40 min at 4 °C. Finally, the slides were transferred in neutralization buffer (100 mM Tris-HCl, pH 8.0) and washed 3 times for 5 min each at 4 °C. 

For the application of the comet assay, after the lysis step nucleoids were digested for 5 min at 37 °C with Fpg (New England Biolabs, Ipswich, MA, USA), diluted 1:3000 in buffer containing 40 mM Hepes (pH 8.0), 0.1 M KCl, 0.5 mM EDTA, and 0.2 mg bovine serum albumin (BSA). After digestion and washing in the same buffer, nucleoids were electrophoresed at 4 °C in denaturation buffer (see above) for 30 min at 25 V. Samples were neutralized in 1.5 M Tris-HCl (pH 7.5) and stained with Hoechst 33,258 dye (0.2 μg/mL).

### 4.4. Immunofluorescence Labeling

Before proceeding with the incubation of the N45.1 monoclonal antibody, acid hydrolysis was performed to denature DNA [39]. After removal of fixative and RNase digestion, the samples were incubated in 2N HCl for 45 min at RT [39]. A neutralization step was then performed with sodium tetraborate 0.1 M (pH 7.8) for 25 min at RT. The cells were then incubated with PBT solution (PBS + 0.2% Tween) containing 1% BSA for 15 min, in order to block non-specific binding of antibodies. After that, the slides were incubated for 1 h at RT in the dark with 50 μL of PBT solution containing 1% BSA together with the N45.1 antibody diluted at 1 mg/mL [39]. Subsequently, cells were washed three times for 10 min each with PBT, and then incubated for 30 min at RT with 50 μL of PBT containing 1% BSA and AF488-conjugated goat anti-mouse secondary antibody (ThermoFisher) diluted 1:200. To improve visualization, a subsequent step of amplification (30 min, RT) was performed with AF488-conjugated donkey anti-goat antibody (Abcam) diluted 1:300. Thereafter, slides were washed 3 times for 10 min each with PBT, then incubated 5 min with a solution of Hoechst 33,258 (0.2 mM) in PBS. After 2 washes in PBS, slides were mounted with Mowiol. 

For labeling of nucleoids or comet samples, higher incubation times (1.5×) of primary and secondary antibodies were used in order to allow thorough diffusion of the reagents.

### 4.5. Avidin-AF488 Labeling

After removing the fixative, the cells were incubated with PBT (PBS + 0.2% Tween) containing 15% FBS for 30 min at RT to avoid non-specific interactions. Then, slides were incubated at RT for 1 h in the dark with 50 μL of PBT containing 15% FBS together with avidin-AF488 (10 mg/mL), as described [32]. At the end of the reaction, the slides were subjected to 3 washes for 5 min with PBT and incubated for 5 min under agitation with a solution of Hoechst 33,258 (0.2 mM) in PBS. After 2 washes in PBS, coverslips were mounted with Mowiol onto slides and observed as described below.

### 4.6. Fluorescence Microscopy and Statistical Analysis

Samples were observed with an Olympus BX51 fluorescence microscope using a 100× oil immersion objective (NA 1.25), and pictures were taken with an Olympus C4040 digital camera. Fluorescence intensity was analyzed with the particle analysis tool of Image J software (version 1.52a, NIH, Bethesda, MA, USA) using DNA images to identify nuclei on which the image with 8-oxodG fluorescent signal was overlapped [40].

Statistical analysis was performed with Prism 6 software (GraphPad, San Diego, CA, USA) used to calculate significance with the Student *t* test (two-tailed), with *p* values < 0.05 considered to be significant. Unless otherwise stated, results are from at least 3 independent experiments.

## 5. Conclusions 

Our results indicate that in situ fluorescence labeling of 8-oxodG with an antibody-based procedure is more reliable than using avidin because of the higher relation to the extent of DNA damage; therefore, it should be preferred for the in situ single cell determination of 8-oxodG. 

Different antibodies that recognize 8-oxodG are commercially available and each one should be evaluated according to possible limitations, including the cross-reactivity with other DNA oxidation products, such as 8-oxoG, 8-oxodA etc. [24]. Interestingly, in U2OS cells treated with 20 and 40 mM KBrO_3_, the monoclonal antibody 15A3 showed an increase in the 8-oxoG fluorescence signal [41] very similar to that observed here. 

Other fluorescence sensors based on genetically modified proteins, small-molecules, or aptamers, have been developed [42,43,44]. However, an in situ analysis of these techniques has been not yet investigated.

In conclusion, the results of our study suggest that detection of 8-oxodG with immunofluorescence provides a reliable determination of this type of DNA damage. 

## Figures and Tables

**Figure 1 molecules-28-04326-f001:**
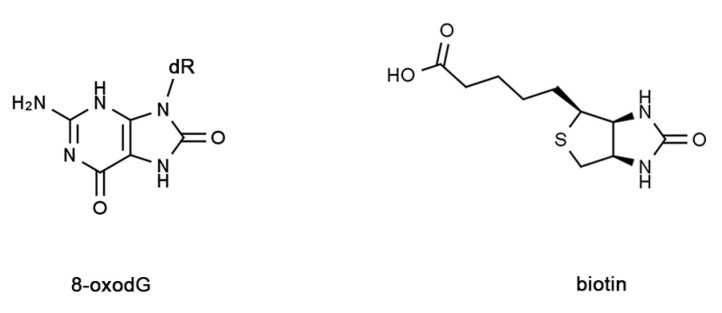
Structural similarity of 8-oxodG with biotin.

**Figure 2 molecules-28-04326-f002:**
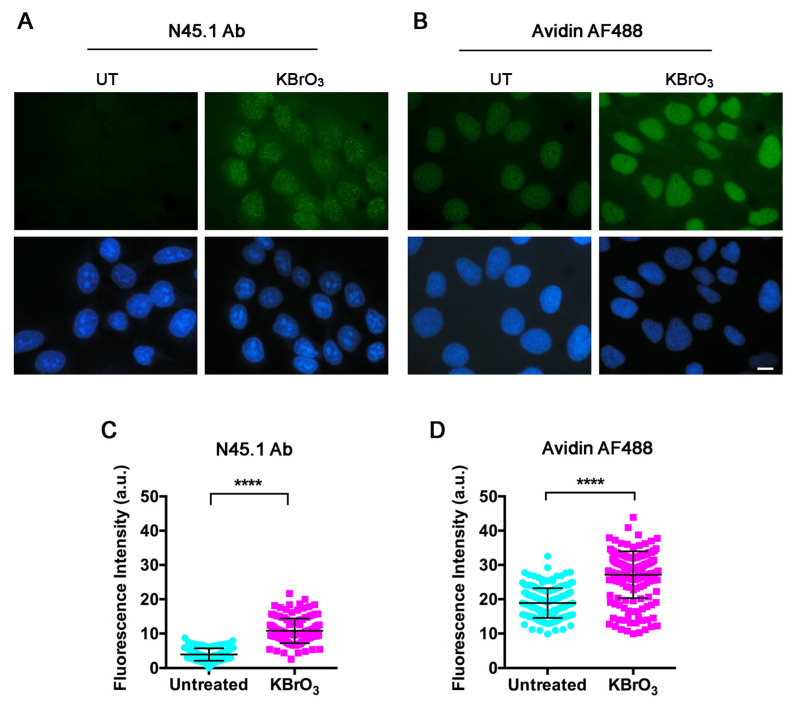
Fluorescence microscopy analysis of 8-oxodG in HaCaT cells untreated (UT) or treated for 30 min with 40 mM KBrO_3_. Detection was performed (**A**) with N45.1 antibody (Ab) or (**B**) with avidin-AF488. Scale bar = 10 μm. (**C**,**D**) Quantification of fluorescence signals from at least 100 cells stained with N45.1 antibody (Ab) or with avidin-AF488. Results are representative from one out of three independent experiments. **** *p* < 0.0001.

**Figure 3 molecules-28-04326-f003:**
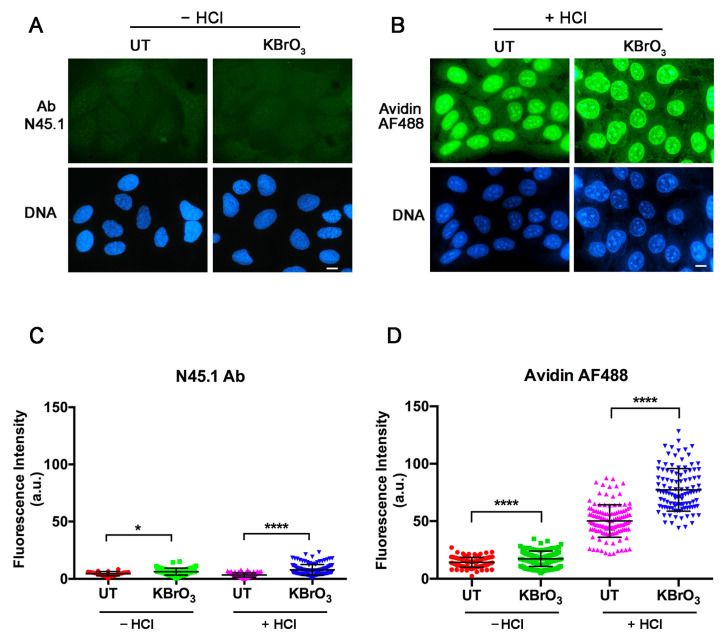
Influence of DNA denaturation on the accessibility of the antibody N45.1 or avidin (AF488-labeled) to DNA in HaCaT cells untreated (UT) or treated for 30 min with 40 mM KBrO_3_. DNA was denatured with 2N HCl for 30 min at room temperature (RT). (**A**) The fluorescence images were obtained after immunofluorescence staining with N45.1 antibody of untreated (UT) or KBrO_3_-treated HaCaT cells in the absence of DNA denaturation (–HCl). In (**B**) the images were obtained from similarly treated and untreated cells processed with avidin-AF488 for 8-oxodG labeling after DNA denaturation (+HCl). In A and B, scale bar = 10 μm. (**C**,**D**) Quantitative analysis of fluorescence intensity of antibody N45.1 (**C**) or avidin-AF488 labeling (**D**) in HaCaT cells untreated (UT) or treated for 30 min with 40 mM KBrO_3_. Samples were exposed (+) or not (–) to HCl for DNA denaturation before labeling reaction. Quantification of fluorescence signals from at least 100 cells for each condition. Results are representative from one out of two independent experiments. * *p* < 0.05; **** *p* < 0.0001.

**Figure 4 molecules-28-04326-f004:**
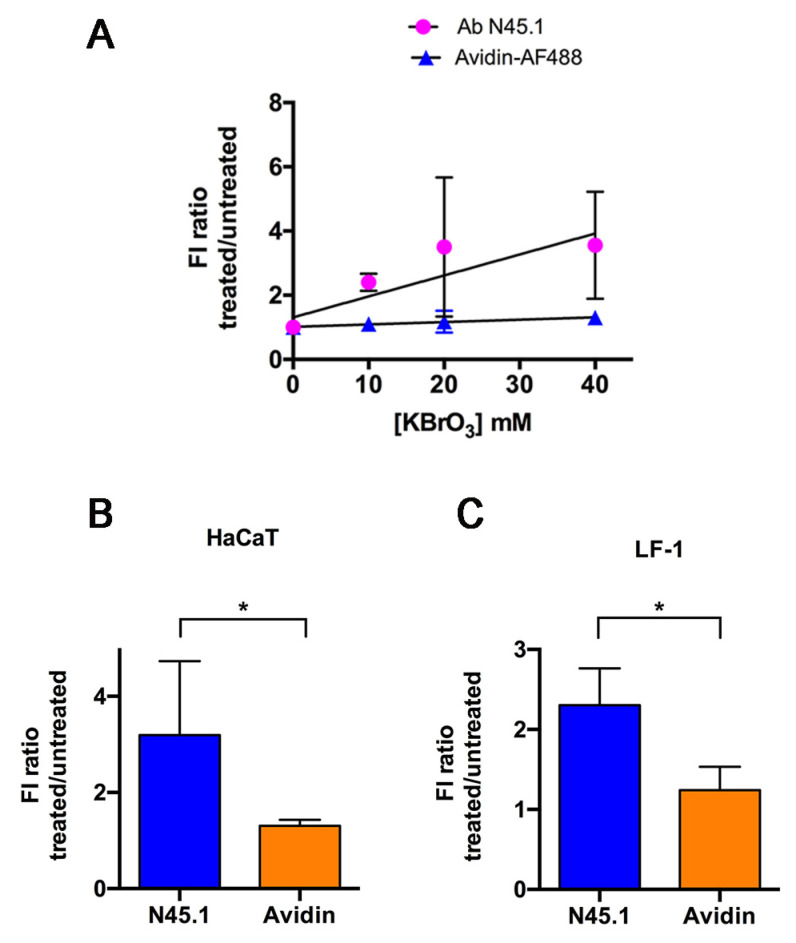
Dependence of fluorescence intensity (FI) of 8-oxodG labeling on the KBrO_3_ concentration. (**A**) HaCaT cells were treated with increasing concentrations of KBrO_3_ for 30 min, then recovered and processed for 8-oxodG detection with the antibody N45.1 or with avidin-AF488, as described in Materials and Methods. Mean values ±S.D. of FI ratio (treated/untreated) obtained from at least three independent experiments. The FI ratio measured in cells treated with 40 mM KBrO_3_/untreated was calculated for labeling of 8-oxodG with N45.1 antibody or with avidin-AF488 in HaCaT cells (**B**) and in LF-1 fibroblasts (**C**). The mean values of the FI ratio from three independent experiments ±S.D. are reported. * *p* < 0.05.

**Figure 5 molecules-28-04326-f005:**
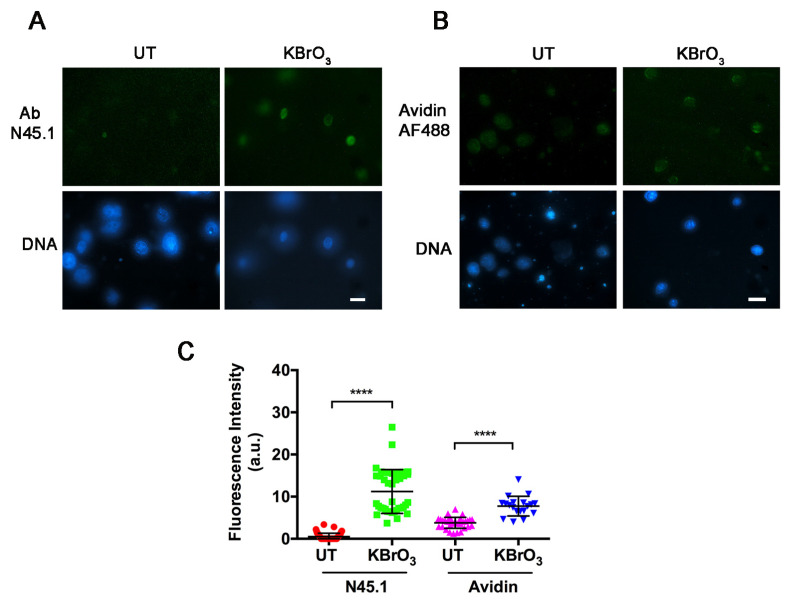
Influence of nuclear protein extraction on 8-oxodG labeling in HaCaT cells treated with 40 mM KBrO_3_ for 30 min or left untreated (UT) and then recovered and processed for nucleoid preparation, as described in Section 4. Nucleoids embedded in agarose gels were processed for 8-oxodG labeling with N45.1 antibody (**A**) or with avidin-AF488 (**B**). Scale bar = 20 μm. (**C**) Quantification of fluorescence intensity of nucleoids obtained from above samples stained with N45.1 antibody or avidin-AF488. At least 25 nucleoids for each condition are shown. Results are from one out of two independent experiments. **** *p* < 0.0001.

**Figure 6 molecules-28-04326-f006:**
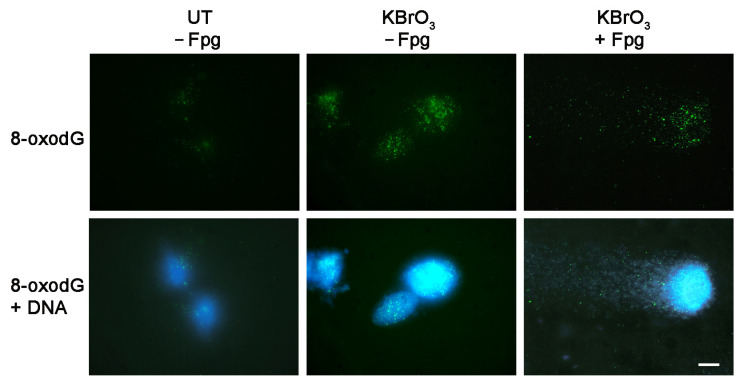
Immunofluorescence labeling of 8-oxodG with N45.1 antibody in SH-SY5Y cells treated with 40 mM KBrO_3_ for 30 min or left untreated (UT) and then recovered and processed for nucleoid preparation, as described in Section 4. Nucleoids embedded in agarose gels were treated with Fpg (+) before electrophoresis. Scale bar = 10 mm.

## Data Availability

The data presented in this study will be available upon reasonable request to the corresponding author.

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
