# Peer review of "Single Cell Determination of 7,8-dihydro-8-oxo-2′-deoxyguanosine by Fluorescence Techniques: Antibody vs. Avidin Labeling"

_molecules, 2023, doi:10.3390/molecules28114326_

Round 1
Reviewer 1 Report
The authors of this manuscript used monoclonal antibody N45.1 and avidin conjugated with the fluorochrome Alexa Fluor488 (AF488) to detect the important biomarker of oxidative damage in cellular DNA (8-oxodG) at the single cell level. They utilized KBrO3 as a chemical inducer of ROS in different type cells. The result indicated that N45.1 showed better performance than avidin-AF488 for the immunofluorescence determination of 8-oxodG. This topic has some importance and the experiment results are good. However, more experiments and discussion can be supplemented and major revision is needed. Some comments are listed as below:
1. What is the detection mechanism of this study? The authors should add a detailed description.
2. What are the possible interference factors and substances? The authors should supplement the selectivity investigation in vitro.
3. What is the detection linear range and the limit of detection for antibody N45.1 towards 8-oxodG?
4. The authors showed that the fluorescence intensity ratio of treated/untreated cells had correlativity with the concentration of KBrO3. What is the corresponding concentration of 8-oxodG? Are there other mature methods to verify the obtained results?
5. The authors used KBrO3 to generate ROS to form 8-oxodG, and this was simulated treatment. What is the actual situation? I wonder if this method can be used for practical detection.
6. The authors should list the relevant references to compare the detection performance of different sensors for detection of 8-oxodG.
7. Conclusion should be set as a separated part to give a detailed summary of this study.
8. The authors should add the photostability results of their fluorescence probes.
Author Response
We thank this Reviewer for the useful comments that we have taken in consideration and relevant modifications have been introduced in the revised manuscript, as specified below.
- The detection mechanism is based on immunofluorescence (antibody) vs the affinity ligand binding, as indicated in the title and in the text. We have now added a new sentence in the Introduction (lines 74-75) of the revised manuscript.
- Possible interference factors and substances (e.g., other oxidation products) for both the antibody and avidin, have been already investigated in the relevant studies performed to characterize the N45.1 antibody (see ref. 24), and that of labeled avidin (see refs. 28 and 38). However, we have added a sentence to highlight the limitations of both methods (see the new paragraph of Conclusions).
- The detection limits for the antibody N45.1 for 8-oxodG, as reported in the original study (ref. 24), are about 10 nM, and the linear range is about between 0.02-0.3 mM 8-oxodG.
- In situ fluorescence determinations do not allow absolute quantification in terms of corresponding concentrations of 8-oxodG. As stated in the introduction, this technique has some advantages when investigating the oxidative DNA damage at the levels of tissues and cells, but are less sensitive when compared with other biochemical methods (refs. 15-18). However, the aim of our study was limited to the comparison of the two in situ techniques, and absolute quantification of 8-oxodG was therefore not possible.
As far as the use of other methods, the fluorescence analysis could be performed with flow cytometry, instead of quantification with Image J. However, flow cytometry will require much more cells and reagents to perform this comparison, and will be not useful for a small-scale analysis, e.g., when cell abundance may be a limiting factor. We have added this consideration in the discussion, which may help to clarify the purpose of our study.
- The use of KBrO3 to generate ROS is quite common and easy to perform (see refs. 33-35, 40). The production of ROS by KBrO3 has been also verified in previous studies (e.g., Zhang et al., Chem-Biol. Interact. 2011, 189: 186-191).
However, to ask the question of the situation induced by KBrO3 treatment in our cell samples, we are now providing more information by showing the analysis of ROS production with the common DCFH-DA assay. The results are shown in the new Supplementary Figure 1.
With the above limitations, the immunofluorescence method can be used for practical detection of 8-oxodG, since this technique is simple, reproducible, and affordable to all laboratories equipped with fluorescence microscopy.
- We have now included (in the Conclusions) recent works describing fluorescent/chemiluminescent sensors which, however, have not been proposed for the “in situ” analysis. Since in our study we have only compared two in situ fluorescence procedures, a complete literature analysis of the performance of the many different sensors used for detection of 8-oxodG, appears to be beyond the aim of this work.
- A “Conclusion” section with a detailed summary of the study has been added to the manuscript.
- In this work we have used commercial reagents using the same fluorochrome to label the secondary antibodies or avidin, i.e., Alexa Fluor 488. This dye is very well-known and previous studies have extensively characterized it, both in the free and in the protein-conjugated form, especially for its optimal photostability properties, as compared with other fluorochromes (e.g., fluorescein).
Reviewer 2 Report
The manuscript entitled: Single cell determination of 7,8-dihydro-8-oxo-2’-deoxyguano- 2 sine by fluorescence techniques: antibody vs avidin labeling; presents an important and interesting study which concluded that the fluorescence labeling of 8-oxodG with an antibody-based procedure is more reliable than using avidin therefore it should be preferred for the single cell determination of 8-oxodG
Materials and methods
-What is the source of HaCaT, Hela and SH-SY5Y neuroblastoma cells
-Include the concentrations of potassium bromate in ‘’Cell treatments’’.
Results:
-Figure 2: 40 mM KBrO3 was used for treatment of cells. Is it possible to include pictures and analysis of one lower and one higher concentration?
Author Response
We thank this Reviewer for the appreciation of our work and for the useful comments that we have taken in consideration and relevant modifications have been introduced in the revised manuscript, as specified below.
Materials and methods
- The source of all cell lines has been now indicated.
- The concentrations of potassium bromate used in the various experiments has been detailed in the “Cell treatments” section.
Results
Figure 2: Examples of pictures taken in cells treated with a lower concentration of KBrO3 (10 mM) is shown in the new Supplementary figure 3 (A and B), in comparison with 40 mM concentration. We have not used concentrations higher than 40 mM because in these conditions, cells will be easily detached due to toxic effects of KBrO3.
Reviewer 3 Report
In this manuscript, XX et al compare avidin-based labelling of 8-oxodG with antibody-based labelling. Their results are impressive, and the manuscript is well-writen. I have minor recommendations, but I do think Figure 3 could use some more attention. As only re-analysis of already collected images (at least, I assume images were collected) is required, I consider this a minor revision. I am looking forward to a revised manuscript, and would be happy to review it again at that stage.
Line 23 – please remove provided by as it is a bit confusing.
Line 28 – if authors can add one sentence on the implication of their findings at the end of their abstract, that would be much appreciated. Later on you talk about use as a biomarker, perhaps something along those lines.
Line 48 – analyzed or described? Please correct if appropriate.
Line 52 – “Several methods for determination of 8-oxoG/dG are in use”.
Line 54 – imply or employ? Please correct if appropriate.
Line 73 – comma after technique is redundant, suggest to remove.
Figure 2 – please provide information about the amount of independently performed experiments this data stems from/is representative of.
Figure 3 – Couple pointers:
1. Please add labels above A and B (antibody/avidin)
2. Please also provide representative images for A and B of the counterpart (either HCl or non-HCl treated, to me it is unclear what A and B are)
3. Perform the analysis in C but then for the antibody labelling (so based on images from dataset A and their (non)HCL-treated counterpart). This way, both techniques can be compared head-to-head.
4. Please provide information about the amount of independently performed experiments this data stems from/is representative of. Please also add the amount of cells quantified.
Figure 5 – please quantify this data.
Line 246, 253 – please describe the source of the HaCaT, HeLa, lymphoblastoid and SH-Sy5Y cell lines.
Line 309 – at what temperature was primary antibody incubation performed at? I’m assuming room temperature? Please add. Same for secondary antibody incubation/amplification antibody incubation.
Line 325 – same question, at what temperature was the avidin labelling performed?
Author Response
We thank this Reviewer for the useful comments that we have taken in consideration and relevant modifications have been introduced in the revised manuscript, as specified below.
- All the suggestions of text modification (lines 23 to 73) have been accepted and introduced in the revised manuscript.
- Figure 2: information on the number of independent experiments has been added in the legend of this Figure and of Figures 3 and 5, as well as in the Statistics section of Materials and Methods.
- Figure 3: We are sorry for the lack of clarity. We have now specifically indicated the experimental conditions of each panel, and have accordingly revised also the text. In addition, quantitative analysis of cells not exposed to DNA denaturation (- HCl) and related to images shown in panels A, has been added. Now in the new figure, quantification is shown side by side, as requested (new panels C and D).
Labels above panels A and B have been added. Information on the number of independent experiments has been also added.
- Figure 5: quantification of nucleoid fluorescence has been performed and added to the figure.
- The source of cell lines has been now indicated in Materials and methods.
- The temperature of incubations with avidin and antibody reactions has been indicated.
Round 2
Reviewer 1 Report
It is OK to accept this manuscript in the present form after the authors addressed my issues and made a revision.
Reviewer 3 Report
Thank you for incorporating my suggestions. Please find one minor thing that can be edited at the typesetting stage:
- The names of 3 authors are in a different font.
Congratulations on a nice paper.